# A Financial Time-Series Prediction Model Based on Multiplex Attention and Linear Transformer Structure

**Caosen Xu** [1] , **Jingyuan Li** [1,*], **Bing Feng** [1] **and Baoli Lu** [2,3]

1   School of Management, Wuhan Institute of Technology, Wuhan 430205, China;
    12006010002@stu.wit.edu.cn (C.X.)
2   Institute of Semiconductors, Chinese Academy of Sciences, Beijing 100083, China
3   School of Computing, University of Portsmouth, Portsmouth PO1 3HE, UK
*   Correspondence: 12006010001@stu.wit.edu.cn

**Abstract:** Financial time-series prediction has been an important topic in deep learning, and the prediction of financial time series is of great importance to investors, commercial banks and regulators. This paper proposes a model based on multiplexed attention mechanisms and linear transformers to predict financial time series. The linear transformer model has a faster model training efficiency and a long-time forecasting capability. Using a linear transformer reduces the original transformer's complexity and preserves the decoder's multiplexed attention mechanism. The results show that the proposed method can effectively improve the prediction accuracy of the model, increase the inference speed of the model and reduce the number of operations, which has new implications for the prediction of financial time series.

**Keywords:** multiplexed attention mechanism; linear transformer; financial time-series prediction

## 1. Introduction

Financial time-series forecasting predicts the future movements of financial markets using time-series data and statistical methods. It can be applied to forecasting in economic areas such as stock prices, currency exchange rates, bond yields, etc. Such forecasting can help investors and financial institutions make more informed investment decisions. Generally, the primary methods for making time-series forecasts in finance are time-series and machine-learning models. Time-series models include the ARIMA [1], VAR [2] and GARCH [3] models. These models can predict future trends based on the direction and pattern of historical data. Machine learning models include BP neural networks [4], support vector machines [5], random forests [6], etc. These models can learn features and practices to predict future trends through training data. However, hybrid models combine multiple methods, such as time-series and machine learning models, and thus have the advantage of combining both benefits to improve their forecasting accuracy.

Data sources for financial time-series forecasting include historical market data, economic data, policy data, etc. To improve the forecasting accuracy, a series of tasks such as data cleaning, feature selection, model training and evaluation is sometimes required [7]. Meanwhile, due to the complexity and volatility of financial markets, financial time-series forecasting faces many challenges, such as data quality problems—historical data may have missing values, outliers and other issues, which require data cleaning and pre-processing; model selection problems—selecting appropriate models and algorithms requires the consideration of data features, forecasting objectives and other factors; overfitting problems—in financial time-series forecasting, there is often the problem of sample data overfitting; model evaluation problems—for different prediction models, it is necessary to make a reasonable evaluation and comparison and choose the optimal model; and forecast error problems—changes in financial markets. Uncertainty makes forecasting

errors unavoidable, so it is necessary to optimize models and algorithms to continuously improve forecasting accuracy.

To better solve the above problems, this paper proposes a financial time-series prediction model based on multiplex attention and a linear transformer based on a transformer [8] that can use distributed CPU for parallel training to improve the training efficiency of the model. When analyzing and predicting longer time series, the semantic association of capturing more prolonged problems is better, and after adding the attention mechanism, the prediction accuracy of the model is higher and the inference speed is faster. Therefore, we use a linear transformer to reduce the complexity of the original transformer and, at the same time, introduce a multiplexed attention mechanism to increase the model's prediction accuracy. The results show that the proposed method has better results in the financial prediction of long time series, which has promising implications for economic time-series prediction. The contribution points of this paper are as follows.

- In this paper, a new transformer model is used to solve the problem of financial time-series forecasting, which is highly innovative, while giving full play to the advantages of the transformer model, simplifying it and introducing the attention mechanism so that it can better solve the problem of financial time-series forecasting.
- Compared with the traditional ARIMA, VAR, GARCH model and other models, it can more accurately predict long time-series data and improve the prediction accuracy. At the same time, it can solve the problem that historical data may have missing values and outliers.
- Compared with machine learning models such as neural networks, support vector machines, or hybrid models such as CNN-LSTM models [9], the proposed method can be trained faster and more efficiently. They can perform more experiments and data analysis and prediction.

In the rest of this paper, we present recent related work in Section 2. Section 3 presents our proposed methods, including an overview, the linear transformer change process and the multiplexed attention mechanism. Section 4 presents the experimental part, details and comparative experiments. Section 5 concludes.

## 2. Related Work

In this section, we selected three models that are commonly used for time-series prediction to represent the different kinds of models, and each of which is briefly introduced in a section below.

### 2.1. GARCH Model

The GARCH model is a standard statistical model used to model changes in volatility in time-series data, which can be used in finance to quantify risk and predict price fluctuations [10]. The GARCH model models current volatility by introducing information about past volatility and considering the autocorrelation and heteroskedasticity of volatility. The model is an extension of the ARCH model [11], where the ARCH stands for autoregressive conditional heteroskedasticity model. In finance, volatility is a fundamental concept because it can be used to quantify the risk of asset prices. The GARCH model is more suitable than the ARCH model to describe volatility in financial markets by establishing the relationship between current and past volatility to predict future volatility and by taking into account the autocorrelation and heteroskedasticity of volatility. However, it also has some limitations, such as being based on basic assumptions and assuming both a standard distribution and a linear relationship [12].

### 2.2. Neural Network Model

The neural network model is an artificial intelligence algorithm inspired by biological neuronal networks that can simulate the information transfer and computational processes between neurons in the human brain. A neural network consists of multiple artificial

neurons connected by weights to form a multilayer structure. The input data are passed and processed through the network to ultimately output the prediction results of the model.

Standard neural network models include multilayer perceptron (MLP), convolutional neural network (CNN) [13] and recurrent neural network (RNN) [14]. Multilayer perceptrons are commonly used to solve classification and regression problems, convolutional neural networks are widely used in areas such as image and speech recognition, and recurrent neural networks are widely used to process sequential data, such as natural language processing and time-series prediction.

The advantages of neural network models include learning nonlinear relationships in the data, their suitability for processing large amounts of high-dimensional data, and the ability to train and optimize end-to-end with back-propagation algorithms. However, neural network models also have some disadvantages, such as requiring large amounts of data and computational resources for training and the poor interpretability of the models [15].

### 2.3. LSTM Model

Hochreiter and Schmidhuber first proposed the long short-term memory network (LSTM) in 1997 as a particular recurrent neural network for processing sequential data with a memory capability and long-term dependent modeling ability, which has achieved excellent performance in many tasks [16]. The core of LSTM is the memory unit, which stores and transmits information. At each time step, the LSTM controls the inflow of input data through an input gate, which information which should be forgotten through a forgetting gate and which information is output through an output gate. This enables LSTM to avoid the gradient disappearance and gradient explosion problems when processing long-sequence data and achieved good performance in areas such as speech recognition, natural language processing and time-series prediction [17]. In addition, there are many variants of LSTM, such as bidirectional LSTM, multilayer LSTM, LSTM with the attention mechanism, etc. These variant models make LSTM better adaptable to different tasks and scenarios and have become a widely used model in deep learning [18].

## 3. Methodology

In this section, we mainly introduce the approach of this paper: firstly, we introduce the overview, then we introduce the replacement process of the linear transformer, the principle of multi-headed attention mechanism and how we combine them.

### 3.1. Overview of Proposed Network

The transformer architecture has a lot of optimization work at this stage, with the main optimization methods being linearization and sparsification. Linear transformer [19] is a variant of the famous transformer architecture widely used for natural language processing (NLP) and other sequence modeling tasks. The main difference between the linear transformer and the original transformer is how the attention mechanism is applied to the input sequence.

Transformer is a deep learning model for natural language processing and other sequence-to-sequence problems, which was proposed by the Google team and released in 2017. Unlike traditional recurrent neural network (RNN) models, the transformer model uses a self-attention mechanism to learn relationships at various locations in the input sequence to better capture long-range dependencies in the series. The model has been widely used for multiple natural languages processing tasks, such as machine translation, language modeling and text generation. The structural flowchart of the transformer is shown in Figure 1 as follows:

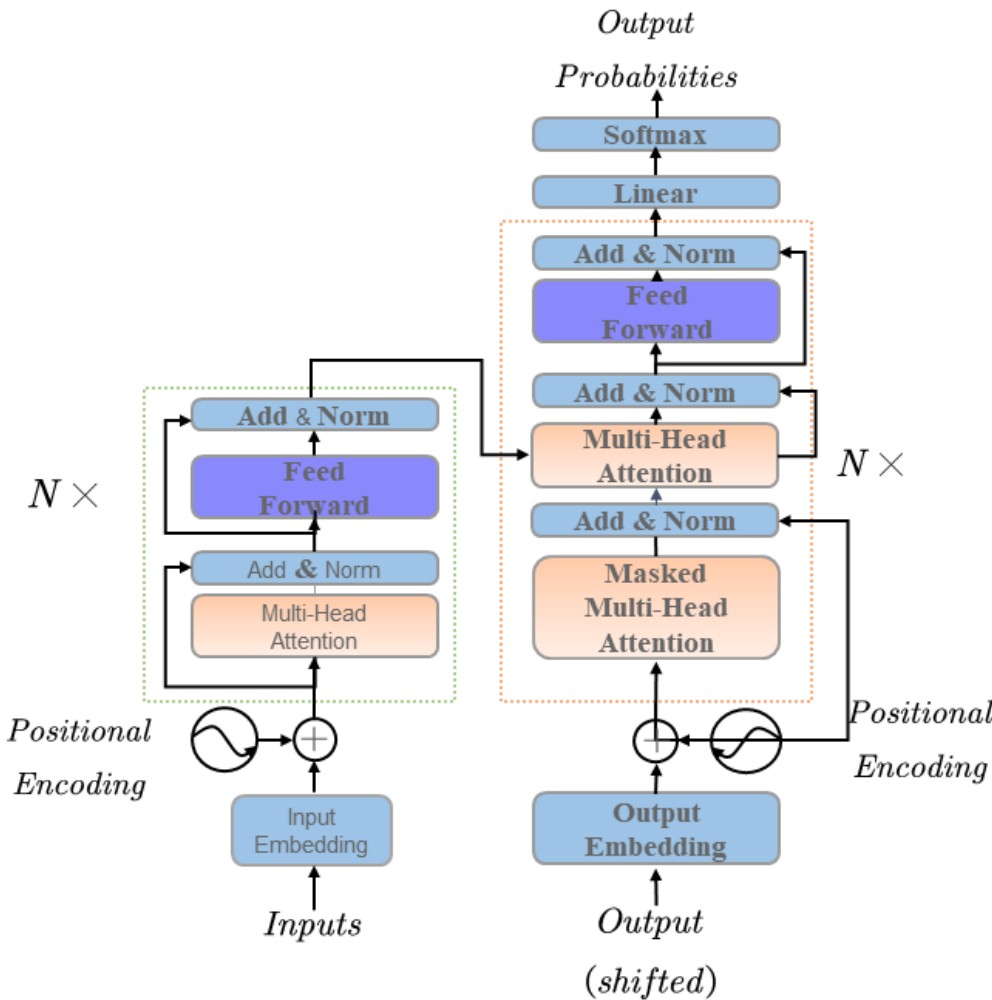

**Figure 1.** Flowchart of the structure of the standard transformer.

The transformer model based on the seq2seq architecture can perform typical tasks in NLP research, such as machine translation, text generation, prediction, etc., while building pre-trained language models for the transfer learning of different jobs. The overall transformer architecture can be divided into four parts: the input part; output part; encoder part; and decoder part [20]. The input contains the source text embedding layer and its location encoder, target text embedding layer and location encoder. The output part has a linear layer and a softmax layer. The encoder part consists of N encoder layers stacked; each encoder consists of two sub-layer connection structures; the first sub-layer connection structure includes a multi-headed self-attentive sub-layer, a normalization layer and a residual connection; the second sub-layer connection structure consists of a feed-forward fully connected sub-layer, a normalization layer and a residual link [21]. The encoder structure is shown in Figure 2.

The decoder part has three sub-layers: the first sub-layer includes a multi-headed self-attentive mechanism and a normalization layer and a residual connection; the second sub-layer includes a multi-headed attention mechanism and a normalization layer and a residual link; the third sub-layer consists of a feed-forward fully connected layer and a normalization layer and a residual connection.

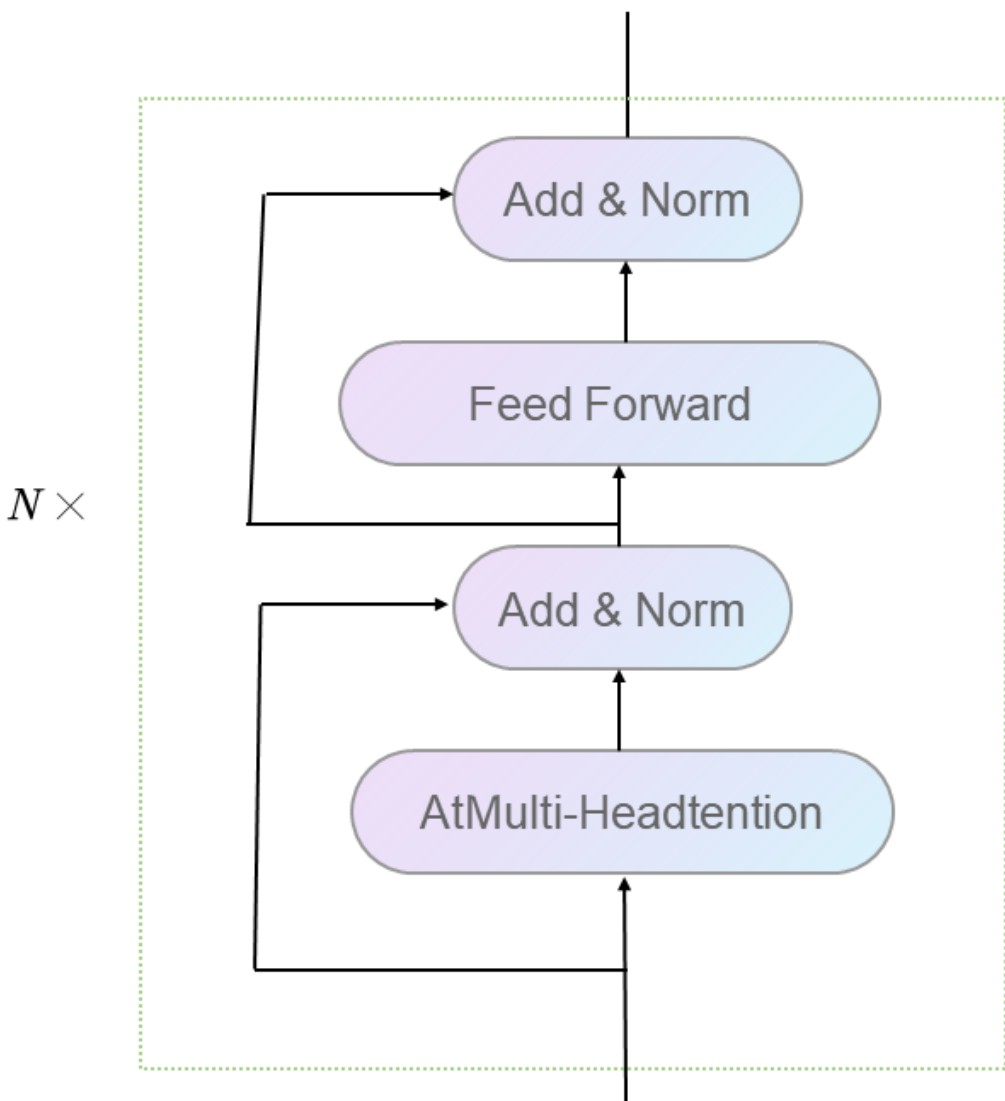

**Figure 2.** Encoder schematic.

### 3.2. Linear Transformer Transformation Process

In a standard transformer, the self-attentive mechanism simultaneously operates on the entire input sequence [22], which can be computationally intensive for long lines. However, in the linear transformer, the input sequence is divided into smaller blocks or segments, and the attention mechanism is applied to each block separately. This approach can handle extended arrangements more quickly and efficiently without sacrificing the modeling capability of the attention mechanism [23]. In this paper, we propose a linearly optimized transformer structure while retaining the essential multiple attention mechanisms in the transformer structure to solve problems stemming from the lack of historical data and accuracy of prediction encountered in the present stage of financial time-series prediction [24]. The matrix operations involved in the encoder's multiple attention mechanism will make the model's time complexity squared. To reduce the complexity of the model and make it better adapted to time-series forecasting, this paper replaces the multiple attention module in the encoder with the linear Fourier transform. Its structure diagram is shown in Figure 3.

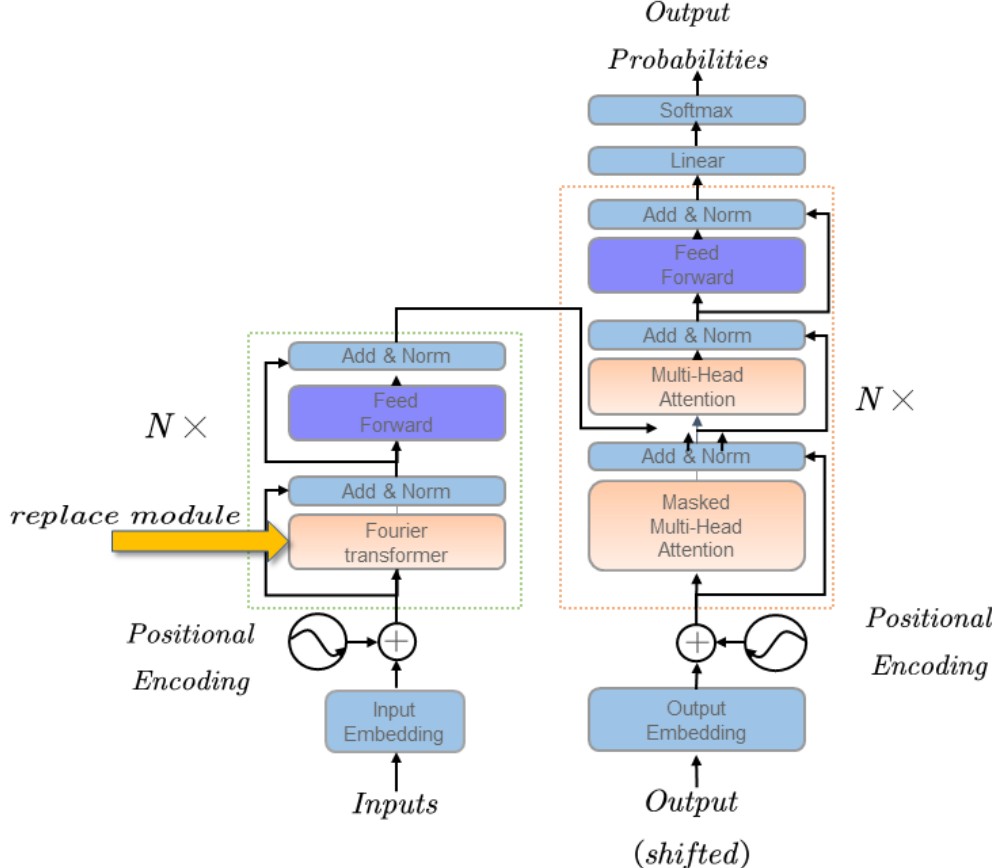

**Figure 3.** Linear transformer structure flowchart.

The replacement Fourier transform equation is as follows. Let the implicit layer dimension be $d_m$ and the sequence length be $N$. In the case of a long sequence input, the self-attentive computation requires the storage space of the $NxN$ attention distribution matrix. The self-attentive module will cause a bottleneck in the model performance [25].

The price series in this paper are discrete price data with hourly and minute frequencies in the time domain and are suitable for the discrete Fourier transform:

$$F(n) = \sum_{t=0}^{N} f(t)e^{-i\frac{2\pi n}{N}t}, \tag{1}$$

$$F(d_s, d_f) = \frac{1}{MN} \sum_{T=0}^{M} \left( \sum_{t=0}^{N} f(T,t)e^{-i\frac{2\pi d_t}{N}t} \right) e^{-i\frac{2\pi d_t}{M}T}. \tag{2}$$

where the input encoding vector dimension of the model is $[\text{Batch-size}, d_s, d_f]$. The Fourier transform is applied to the sequence length dimension $d$. It is equivalent to decomposing the sequence into multiple sinusoidal superposition forms in this dimension; then, the Fourier transform is performed on the feature variable dimension $d_t$, which is again decomposed into sinusoidal superposition forms in the feature variable dimension.

We use the two-dimensional Fourier transform to transform the time-domain sequence into the frequency domain and perform feature extraction based upon it. Compared with the multi-headed self-attentive mechanism, this paper adopts a linear transformer using the Fourier transform module. This linear structure, without introducing parameters, does not increase the complexity of the model but also reduces the model complexity to the linear level [26].

In this paper, the decoder part of the transformer is replaced and the replacement process is shown in Figure 4.

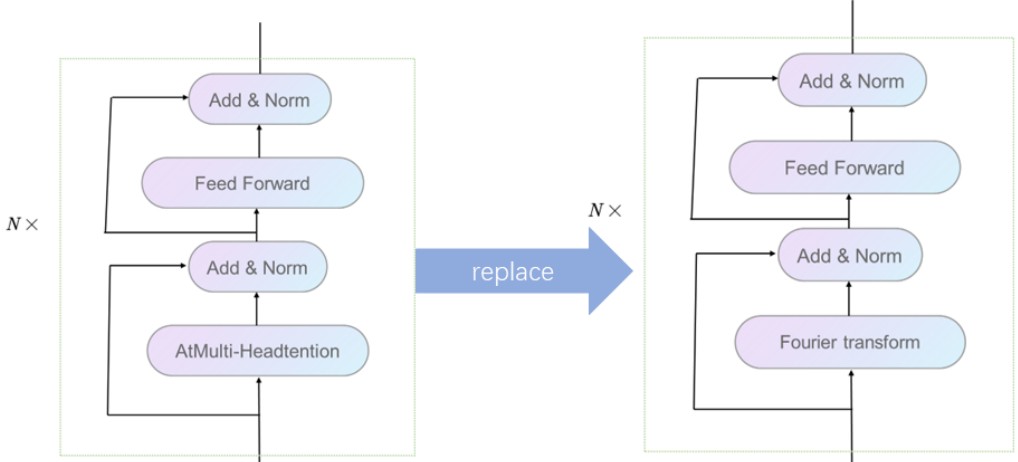

**Figure 4.** Linear transform replacement process.

Except for the Fourier transform, the formulas for calculating the remaining parts of the linear transformer, which are the same as the standard transformer, are as follows.

(1)  Self attention:

$$\text{Attention}(Q, K, V) = \text{softmax}\left(\frac{QK^T}{\sqrt{d_k}}\right)V, \tag{3}$$

where $Q$ is the query matrix; $K$ is the content to be attended; $QK^T$ is the dot product operation, which calculates the attention weight for $Q$ on $V$; the purpose of scaling by $\sqrt{d_k}$ is to avoid an excessively large dot product because the gradient after *softmax* will be slight when the dot product is excessively large. Another advantage of *softmax* is that it facilitates the back-propagation gradient calculation while smoothing the results to the 0–1 interval. The initial $Q, K, V$ are the same, resulting from the sum of word embedding and position embedding.

It is worth noting that $K$ and $V$ in the second layer of the decoder are from the encoder, and $Q$ is from the output of the first layer of the decoder. This is the reason why the Decoder stage cannot be parallelized.

(2)  Position-wise, feed-forward networks:

$$FFN(x) = \max(0, xW_1 + b_1)W_2 + b_2. \tag{4}$$

The location fully connected to the feed-forward network provides nonlinear transformations for the two *Dense* layers via the *Relu* activation function. Position-wise means that the input and output dimensions are the same. For example, the size of the input/output is $d_{model}$ = 512 and the size of inner fully connected layer is $d_{ff}$ = 2048.

### 3.3. Multiplexed Attention Mechanism

A multiplex attention mechanism in a deep learning model is an integral part of the transformer structure that adaptively selects different subsets of input features to compute the output [27]. Specifically, the multiplex attention mechanism can simultaneously consider different feature subspaces and add a specific weight for each subspace. These weights can be used to perform a weighted average of the features in each feature subspace to produce a summary feature representation. Eventually, the multiplexed attention mechanism combines multiple feature representations and passes them to the next layer of the network for processing—the structure of the multiplexed attention mechanism is shown Figure 5.

The formula for calculating multiplexed attention is as follows.

$$\text{MultiHead}(Q, K, V) = \text{Concat}(\text{ head }_1, \ldots, \text{ head }_h)W^O, \tag{5}$$

where:

$$\text{head }_i = \text{ Attention}\left(QW_i^Q, KW_i^K, VW_i^V\right). \tag{6}$$

In the formula $W_i^Q \in R^{d_{\text{model}} \times d_k}$, $W_i^K \in R^{d_{\text{model}} \times d_k}$, $W_i^V \in R^{d_{\text{model}} \times d_v}$, $W_i^O \in R^{hd_v \times d_{\text{model}}}$, if $h = 8$, then $d_k = d_v = d_{\text{model}}/h = 64$.

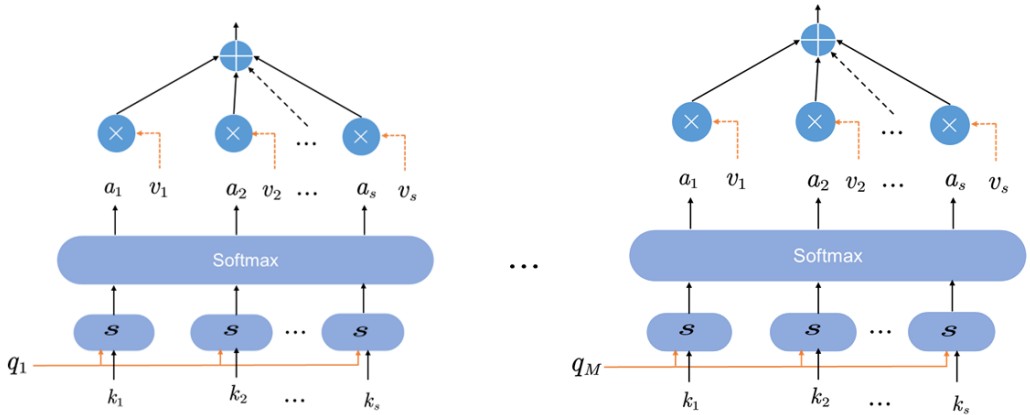

**Figure 5.** Multi-way attention schematic.

Each head computes its attention so that parallel computations can be performed simultaneously and different aspects of attention can be learned in different subspaces. They are then stitched together and multiplied by $W_O$ to obtain the final feature representation.

Multiplexed attention mechanisms perform well in many deep learning tasks, especially those that require the simultaneous consideration of multiple input sources. For example, in a natural language processing task, a multiplexed attention mechanism can simultaneously consider different words in a sentence to better capture their semantic relationships. In image processing tasks, multiple attention mechanisms can view the other regions of an image simultaneously and extract important features.

In the following, the training process Algorithm 1 of this paper is described in detail. In the standard transformer, the multi-headed attention in the encoder is replaced by the Fourier transform to achieve the effect of dimensionality reduction, making the transformer structure more usable for financial time-series prediction and reducing the number of operations and training time.

---

**Algorithm 1:** Algorithmic representation of the training process in this paper.

Training set D, validation set $V$, learning rate $\varphi$, number of network layers $L$.
Input: self-attention
Encoder: the Fourier transform, feed-forward networks
Decoder: multiplex attention mechanism, feed-forward networks

**while** *The error rate of the neural network model on the validation set vs. no longer decreases* **do**

    Randomly reorder the samples in the training set $D$

    **for** $n = 1, \cdots, N$ **do**

        Select samples from the training set$(x(n), y(n))$ Feed-forward networks calculation of net inputs$(l)$ for each layer until the last layer
        Backpropagation calculates the error $\delta(l)$ of each layer Calculate the derivative of each layer parameter Update

    **end**

**end**

---

## 4. Experiment

In this section, the main introduction is the experimental data setup, the details of the experiment and the experimental results and analysis.

### 4.1. Datasets

The data for this paper were mainly obtained from the National Bureau of Statistics of China, CSMAR [28], IMF [29], Citigroups, Redhat Software (2021-11-09 RHSA-2021:4356 4.18.0-348) and the Standard & Poor500 stock index [30].

The National Bureau of Statistics of China (NBSC) is the Chinese government department responsible for national economic and social statistics. Its primary responsibilities are formulating guidelines, policies and standards for statistical work, organizing and implementing national statistical surveys and reports, publishing national economic and social statistics and analyzing and interpreting data. Its statistics have essential reference value for the government in formulating financial and social development plans and making policies and decisions. At the same time, the data it publishes are widely followed and cited by scholars, enterprises and media at home and abroad. The National Bureau of Statistics of China was established in 1952 and is headquartered in Beijing. It has several departments and agencies under it, including the National Data Center, the National Statistics Information Center and the National Computer Information Center. Its statistics have financial data available for use in this paper [31].

CSMAR is China's financial and economic database, providing comprehensive data and research services to academic and professional researchers, government agencies and businesses. It covers various topics, including stocks, bonds, futures, options, funds, macroeconomics, industry analysis, corporate governance and social responsibility. CSMAR is widely used in academic research, investment management, risk management and financial regulation. It provides access to various financial and economic data, including historical and real-time market data, company financial statements, analyst reports and economic indicators. Its services include data cleansing, mining and analysis tools that enable researchers to efficiently process large datasets. CSMAR's clients include academic institutions, investment banks, asset management firms, government agencies and other organizations that require accurate and comprehensive financial and economic data. Researchers use CSMAR to investigate various topics, such as asset pricing, market efficiency, corporate finance and financial regulation. Investors use CSMAR to analyze market trends, assess investment opportunities and manage risk. Regulators use CSMAR to monitor market activity, detect fraud and develop policies that promote financial stability and investor protection [32].

The International Monetary Fund (IMF) is an international organization that promotes international monetary cooperation, facilitates international trade, promotes exchange rate stability and provides resources to help countries needing financial assistance. The IMF was founded in 1944 and is headquartered in Washington, DC. The IMF's primary purpose is to ensure the stability of the international monetary system, which includes exchange rate stability and the promotion of balanced economic growth. The IMF provides financial assistance to member countries experiencing the balance of payment problems or other economic difficulties to achieve this goal. The IMF also offers financial advice and technical assistance to its member countries and researches global economic issues. The organization is governed by its member countries, which have the authority to make decisions regarding the policies and operations of the IMF. The IMF plays an essential role in the global economy by promoting economic stability and assisting countries in need [33].

Citigroup is a global financial services company that provides a broad range of financial products and services to consumers, businesses, governments and institutions. The company was founded in 1998 through the merger of Citigroup and Travelers Group and is headquartered in New York City. Citigroup operates in two primary divisions: Global Consumer Banking and Institutional Clients Groups. The Global Consumer Banking segment provides a range of banking and financial services to consumers and small businesses, including credit cards, mortgages and personal loans. The Institutional Clients

Group offers various financial services to corporations, governments and institutions, including investment banking, capital markets and corporate banking. Citigroup operates in over 100 countries and serves over 20 billion customer accounts. With more than 200,000 employees worldwide, the company is one of the world's largest financial institutions. Citigroup is also known for its philanthropic efforts, including the Citigroup Foundation, which supports initiatives in education, financial inclusion and sustainable development [34].

We took two indicators, the stock's opening and closing prices, as our predictor, and selected data from four datasets during a specific period for training and prediction. The selected data are shown in Table 1 below.

**Table 1.** The time period selected for each dataset.

| Dataset | Time Periond | Code |
|---------|--------------|------|
| CSMAR | Code 01-01-2010 to 01-01-2022 | Dataset 1 |
| NBS | Code 24-11-2010 to 24-11-2022 | Dataset 2 |
| IMF | 01-11-2011 to 01-11-2022 | Dataset 3 |
| Citigroups | 01-01-2012 to 01-01-2022 | Dataset 4 |
| Redhat | 01-02-2001 to 01-02-2022 | Dataset 5 |
| S&P | 50015-05-2004 to 15-05-2022 | Dataset 6 |

*4.2. Experimental Details*

In this paper, based on the characteristics of the transformer, we first compared the training time of different models with the proposed model. We also performed experiments on more complex data. We compared the prediction accuracy of other models on different datasets to check the generalization of the proposed model, and we also compared the prediction accuracy of different models on more complex datasets. Then, we compared the model's inference speed and AUC to test the performance of the proposed model.

*4.3. Experimental Results and Analysis*

In Figure 6, we compared the training time of LSTM [35], transformer [36], SVM model [37] and the proposed model at different inference numbers to check the training time of the proposed model. The table shows no significant difference between several models at 200 or 400 inference numbers. Still, at 600 or 800 inference numbers, the inference speed of the proposed model is significantly lower compared to other models, which indicates that the advantage of the proposed model in terms of training time becomes more evident as the inference number increases.

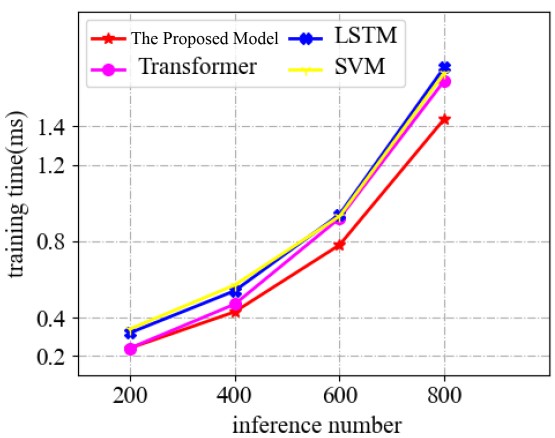

**Figure 6.** Comparison of efficiently training time of different models.

In Figure 7, we compare the training time variation of LSTM, ARIMA [38] and BP network models [39] and on more complex data. Financial data are often more difficult, diverse, highly variable and subject to various factors. Therefore, the ability to perform well in more complex datasets is an essential indicator of a financial time-series forecasting model. The experimental results show that the proposed model has a significant advantage over other models in terms of training time on complex datasets, thanks to the distributed training structure of the linear transformer.

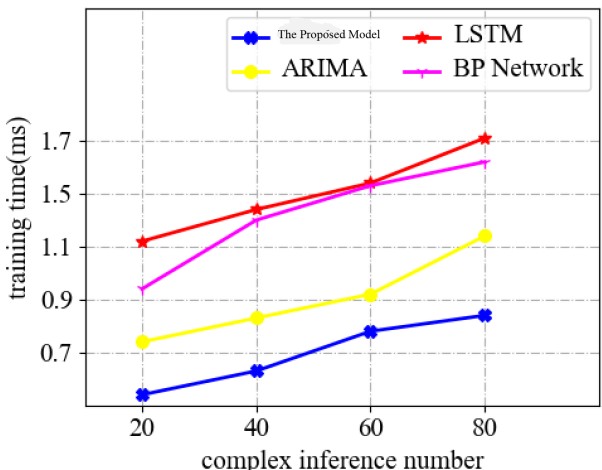

**Figure 7.** Comparison of the training time in complex datasets.

In Figure 8, we compare the prediction accuracy of the proposed model with ARIMA and BP network models for financial time series in cases with different levels of complexity; the prediction accuracy of a model is an essential indicator of a model, so we first selected three models for a comparison of the prediction accuracy.

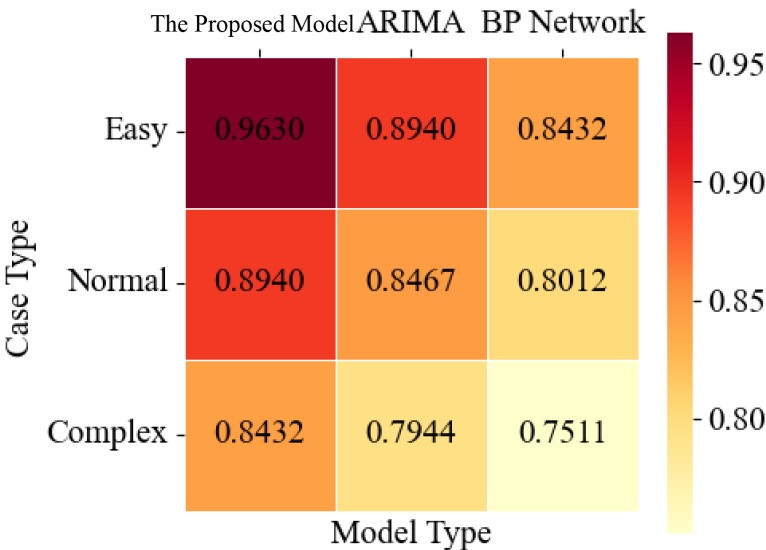

**Figure 8.** Comparison of the prediction accuracies of different models.

In Figure 9, we compare the number of parameters of the transformer, LSTM and SVM models with the proposed model. To more intuitively represent the number of parameters of our linear transformer compared with the standard transformer structure, we selected a histogram for comparison, from which we can see that the number of parameters of the proposed model is significantly lower than that of the standard transformer model, which also has some advantages over the LSTM and SVM models.

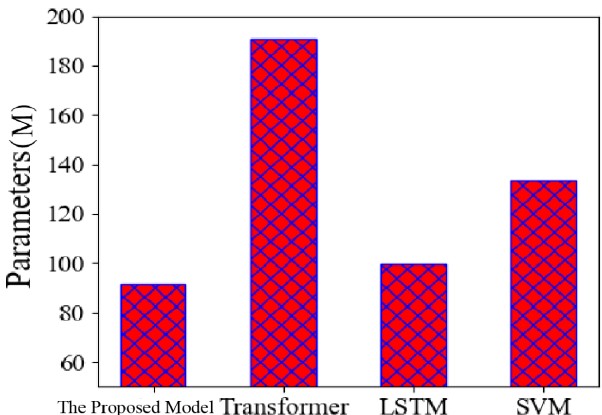

**Figure 9.** Comparison of the parametric quantities of models.

The smaller the calculation amount of a model, the simpler the calculation process of the model and the faster the calculation speed of the model, which is one of the important criteria to measure a model. In Figure 10, we compare the calculation volume of different models. A comparison is made, and the experimental results show that the computational load of the proposed model is significantly lower than in other models, especially in comparison with the transformer model, the advantages of the proposed model are more obvious [40].

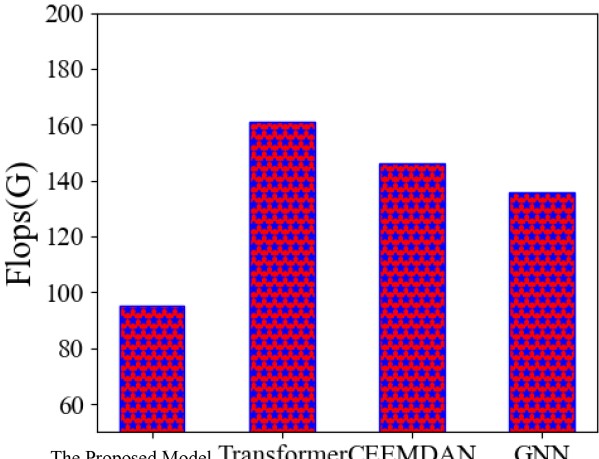

**Figure 10.** Comparison of the amount of operations in different models.

In Figure 11, we compare the inference speed size of the model on four datasets, namely Redhat, Standard & Poor500 stock index, CSMAR and NBS, by first determining the number of inferences and then recording the inference time in different data from the figure. We can see that, compared with other models, our linear transformer model has a more significant advantage in inference speed and a better overall performance.

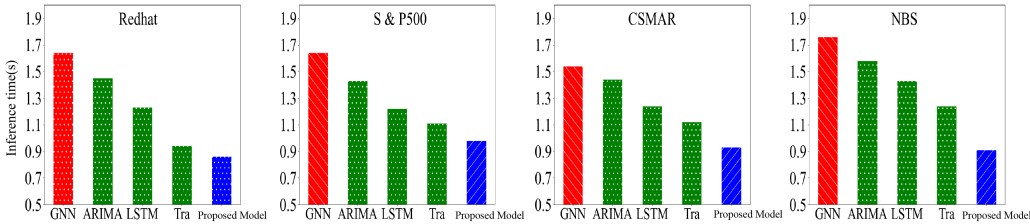

**Figure 11.** Comparison of inference speed for different datasets.

In Figure 12, to further compare the prediction accuracy of the proposed model, we selected data from NBS, CSMAR, IMF and Citigroups, four different datasets for the experiments, and three other more competitive models, namely the transformer, LSTM and GNN models [41], which also perform well in time-series prediction. Through the experimental comparison, our linear transformer model has an excellent performance in different datasets, which can effectively prove the proposed model's prediction accuracy and generalization to other data types' scenarios.

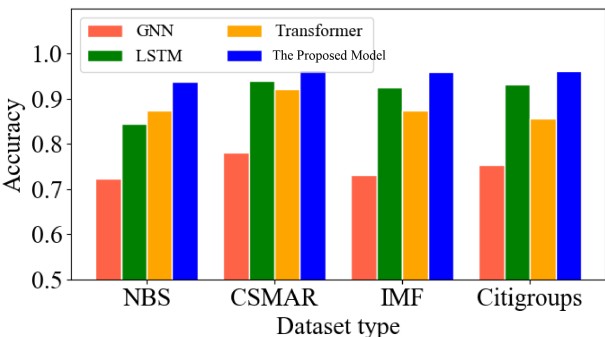

**Figure 12.** Accuracy in different datasets.

As shown in Figure 13, to further validate the prediction accuracy of the proposed model for financial timing, we compare the AUC values of the proposed model and the LSTM model, and the results show that the proposed model performs better than the LSTM model for different data groupings, all of which are from the NBS database. This effectively measures the prediction accuracy of the proposed model. The area under the ROC curve (AUC) is a commonly used binary classification model evaluation index, whose value ranges from 0 to 1. The larger the AUC, the better the model's performance and ability to distinguish between positive and negative samples.

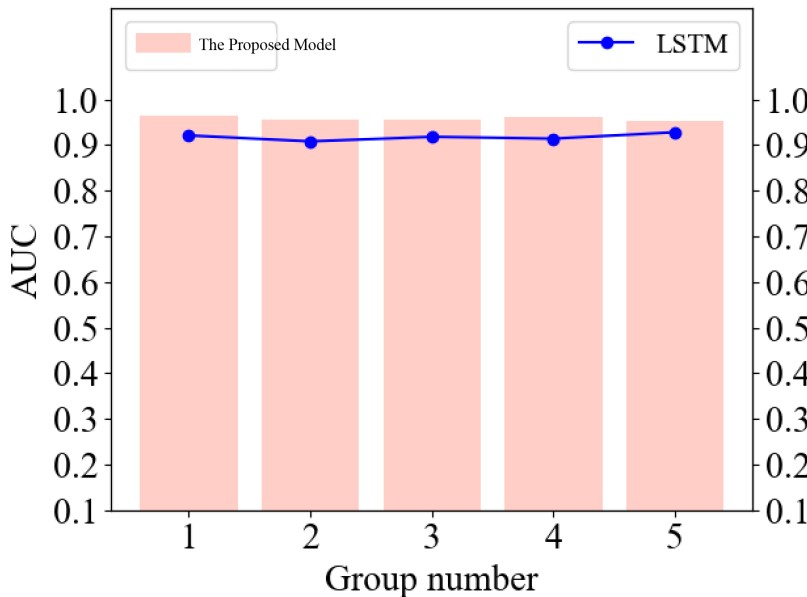

**Figure 13.** Comparison with the AUC of LSTM model.

In Table 2, we summarize the models compared in our experiments, as well as the three most intuitive metrics, namely the prediction accuracy, the number of parameters and the number of operations, to allow us to more intuitively compare the performance of the proposed model. The sources of data for each model are also summarized.

**Table 2.** A comparison of the different models.

| Model | Accuracy | Flops (G) | Parameters (M) |
|---|---|---|---|
| CEEMDAN [40] | 0.931 | 146.33 | 178.43 |
| GNN [41] | 0.780 | 135.87 | 179.99 |
| SVM [37] | 0.823 | 102.50 | 133.43 |
| BP Network [39] | 0.843 | 86.34 | 137.25 |
| ARIMA [38] | 0.894 | 150.66 | 178.27 |
| LSTM [35] | 0.921 | 112.43 | 99.86 |
| Transformer [36] | 0.940 | 160.88 | 190.54 |
| **Ours** | **0.963** | **95.32** | **91.45** |

## 5. Discussion

Financial time-series forecasting is essential in the financial market. The linear transformer model we proposed based on the multi-head attention mechanism is used to predict financial time series. This can help investors and financial institutions make more informed decisions, reduce risks and obtain higher returns. It plays a more critical role in the prosperity and stability of the financial market.

Financial timing forecasting is essential to investors and financial institutions because it helps them make more informed decisions, reduce risk and achieve higher returns. Economic time-series forecasting serves several purposes: risk management—financial time-series forecasting can help investors identify and avoid risks, as investors can develop better risk management strategies by forecasting market trends and price changes; investment decisions—financial time-series forecasting can help investors make more informed investment decisions, as they can choose financial products such as stocks, foreign exchange, commodities or bonds for investment based on the forecasting results; business decisions—economic time-series forecasting can also help financial institutions make better business decisions, as financial institutions can adjust their business strategies and product portfolios to meet customer needs and market changes based on forecasting results; asset pricing—financial timing forecasting can help financial institutions more accurately price financial products as they can better assess the value of their assets and liabilities by forecasting market trends and price changes; financial planning—financial timing forecasting can help investors and financial institutions make better financial plans as they can make better budgets and investment plans based on the market forecasting results

## 6. Conclusions

In this paper, we propose a financial time-series prediction model based on multiplexed attention and a linear transformer structure for economic time-series prediction to solve some problems of the missing historical data of traditional models and machine learning models in long time-series prediction and to improve the training rate and accuracy of financial time-series prediction in machine learning. Since the traditional transformer structure is complicated, in this paper, we simplify the transformer structure, replace the original transformer structure with the linear transformer and replace the multi-headed attention module in the encoder part of the transformer set structure with a linear Fourier transform. To make the transformer more applicable to the prediction of financial timing while retaining the multi-headed attention part in the decoder, we can improve the model's training speed and prediction accuracy based on our work. At the same time, due to the linear transformer structure, the number of operations and the number of parameters of the proposed model are reduced compared with the standard transformer model, which significantly improves the computational speed of the model. In the field of financial time-series forecasting, this occurs by specific revelation.

Linear transformer models based on multi-headed attention mechanisms are highly effective in various natural language processing tasks, including language modeling, machine translation and text classification. However, they also have some limitations and potential directions for future research. One limitation of linear transformer models is that they can be computationally expensive, especially when dealing with long time-series sequences. This can slow training and inference times, which can be a problem in real-time applications. To address this, researchers are exploring ways to optimize the computation of attention weights, such as using scant attention or pruning attention heads. Another limitation is that linear transformers must be better suited for tasks requiring reasoning or inference, such as question-answering or dialogue generation. This is because they lack explicit mechanisms for representing and manipulating structured knowledge. Researchers are exploring ways to incorporate external knowledge sources, such as knowledge graphs or ontologies, into the model architecture to address this. In terms of future directions, one promising area of research is the development of hybrid models that combine the strengths of linear transformers with other neural network architectures, such as convolutional neural networks or graph neural networks. Another direction is the exploration of unsupervised pre-training techniques, such as generative pre-training or contrastive learning, to improve the efficiency and effectiveness of linear transformer models.

**Author Contributions:** Conceptualization, J.L. and C.X.; methodology, B.F. and B.L.; software, J.L., C.X. and B.F.; validation, C.X., B.F. and B.L.; formal analysis, C.X., B.F. and B.L.; investigation, C.X., B.F. and B.L.; data curation, C.X. and B.F.; writing—original draft preparation, J.L., C.X. and B.L.; writing—review and editing, J.L. and B.F.; visualization, J.L., C.X. and B.F.; supervision, J.L.; funding acquisition, B.L. All authors have read and agreed to the published version of the manuscript.

**Funding:** This work is supported by National Key R&D Program of China (2020AAA0105200).

**Institutional Review Board Statement:** Not applicable.

**Informed Consent Statement:** Not applicable.

**Data Availability Statement:** Not applicable.

**Conflicts of Interest:** The authors declare no conflict of interest.

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
