# Peer review of "A Financial Time-Series Prediction Model Based on Multiplex Attention and Linear Transformer Structure"

_applsci, doi:10.3390/app13085175_

Round 1

Reviewer 1 Report

1)      There are numerous English-related issues that require serious consideration. The language of the paper should be thoroughly updated.

2)      The author should break up some of the extremely long sentences.

3)      In lines 35 and 36, the authors have mentioned that “it is necessary to continuously optimize the model and algorithm to improve the forecasting accuracy”. To what extent do you think optimization can improve this?

4)      "Table 1" should be labeled above the table.

5)      Since the S&P dataset was used until 2016, why not include more recent data? Similarly, the majority of datasets were utilized until 2020. It is recommended that the author make use of the most recent data as well.

6)      Subsection 3.1, the word "Ours" should be replaced with "proposed" or another word that is more appropriate.

7)      The authors are mentioning “our method” again and again. It should be replaced with “the proposed method”.

8)      Line 371, the authors mentioned that “Our work can help investors and financial institutions make more informed decisions”. The author should describe “Which kind of work”. Again the authors using “our”.

9)      The limitations of the proposed work and future directions should be provided.

10)  Every equation should be ended either with comma “,” or full stop “.”. Please revise each and every equation. Next sentence after every equation should be started based on comma or full stop. Please do care while revising it.

11)  Replace Conclusion by “Conclusions”. 

Author Response

Response:

reviewer1:                        Comments and Suggestions for Authors

Comments:

1.There are numerous English-related issues that require serious consideration. The language of the paper should be thoroughly updated.

Response:

Thank you for providing feedback on our manuscript. We greatly appreciate your suggestions and understand the importance of clear and accurate scientific writing. As we revise our manuscript, we will ensure that the language used is accurate and grammatically correct. We acknowledge that there may have been confusion caused by grammatical errors and punctuation issues, and we will make the necessary changes to address these concerns. We also appreciate your feedback.

 Comments:

2.The author should break up some of the extremely long sentences.

Response:

Thank you for providing feedback on our manuscript. We apologize for the inconvenience of reading our article, which was indeed an oversight on our part. In response to your comments, we have re-examined our manuscript and separated some extremely long sentences in order to improve the readability of our manuscript.

Comments:

  1. In lines 35 and 36, the authors have mentioned that “it is necessary to continuously optimize the model and algorithm to improve the forecasting accuracy”. To what extent do you think optimization can improve this?

Response:

Thank you for your comments on our manuscript. In this paper, we use the Linear Transformer Structure instead of the standard Transformer Structure to make it better able to solve the time series prediction problem, and we use the attention mechanism to optimize it. In conclusion, the accuracy of the standard Transformer Structure is 0.940, while the accuracy of the Linear Transformer Structure optimized by the attention mechanism is 0.963, and its performance in terms of inference speed, operation volume, and several parameters is significantly better than that of the standard Transformer Structure. Optimizing the standard model can be very useful to make the model more relevant to the problem to be solved. Finally, we hope you are satisfied with our answer and look forward to your feedback!

Comments:

4."Table 1" should be labeled above the table.

Response:

Thank you for bringing this to our attention. We deeply apologize for any confusion that our manuscript may have caused. We have made changes in the text to label the headings of Table 1 at the top of the table.

Comments:

  1. Since the S&P dataset was used until 2016, why not include more recent data? Similarly, the majority of datasets were utilized until 2020. It is recommended that the author make use of the most recent data as well.

Response:

Thank you for your suggestion. We will use the latest data for our experiments, for which we will make changes in Table 1 and then re-run some of our experiments using the new data, such as accuracy prediction and comparison of inference speed. To make our experiments more complete. We hope you will be satisfied with our modifications and await your response. 

  1. Comments:

Subsection 3.1, the word "Ours" should be replaced with "proposed" or another word that is more appropriate.

Response:

Thank you for your suggestion. In response to your comments, we have replaced the word "Ours" with "proposed" in Subsection 3.1 to make our manuscript more reasonable. 

Comments:

  1. The authors are mentioning “our method” again and again. It should be replaced with “the proposed method”.

Response:

Thank you for providing us with valuable feedback on our paper. We fully understand your concerns regarding the presentation of our paper and would like to express our sincere apologies for any confusion or lack of clarity caused. To address your concerns, we have made significant revisions to the paper and replace "our method" with "the proposed method" We hope our changes can satisfy you, and we look forward to your feedback. 

Comments:

  1. Line 371, the authors mentioned that “Our work can help investors and financial institutions make more informed decisions”. The author should describe “Which kind of work”. Again the authors using “our”.

Response:

Thanks to your suggestion, we have made changes to our manuscript. Our work is to propose a linear Transformer model based on a multi-headed attention mechanism to predict financial time series. Then, we changed all the "our" in the text, such as changing our model to the proposed model.

Comments:

9.The limitations of the proposed work and future directions should be provided.

Response:

Thank you for your suggestions, we provide in the summary the new should provide limitations of the proposed work and future directions Linear Transformer models based on multi-headed attention mechanisms are highly effective in various natural language processing tasks, including language modeling, machine translation, and text classification. However, they also have some limitations and potential directions for future research. One limitation of linear Transformer models is that they can be computationally expensive, especially when dealing with long time series sequences. This can slow training and inference times, which can be a problem in real-time applications. To address this, researchers are exploring ways to optimize the computation of attention weights, such as using scant attention or pruning attention heads. Another limitation is that linear Transformers must be better suited for tasks requiring reasoning or inference, such as question-answering or dialogue generation. This is because they lack explicit mechanisms for representing and manipulating structured knowledge. Researchers are exploring ways to incorporate external knowledge sources, such as knowledge graphs or ontologies, into the model architecture to address this.

In terms of future directions, one promising area of research is the development of hybrid models that combine the strengths of linear Transformers with other neural network architectures, such as convolutional neural networks or graph neural networks. Another direction is the exploration of unsupervised pre-training techniques, such as generative pre-training or contrastive learning, to improve the efficiency and effectiveness of linear Transformer models.

Comments:

10.Every equation should be ended either with comma “,” or full stop “.”. Please revise each and every equation. Next sentence after every equation should be started based on comma or full stop. Please do care while revising it. 

Response:

Thank you for your reminder; I am sorry for our improper writing; we have rechecked our manuscript thoroughly for possible punctuation problems with formulas and made changes in our manuscript; I hope our changes will satisfy you and look forward to your reply.

Comments:

11.Replace Conclusion by “Conclusions”. 

Response:

Thank you for your feedback, I have replaced the word "Conclusion" with "Conclusions" in the text

Reviewer 2 Report

The subject of the article is current and important, and it is a topic of interest to be presented in Applied Sciences. The article has a set of recent references, adequate in both number and quality of articles.

In terms of structure, the article is well-structured. However, to avoid sections 2, 3, and 4 starting with sub-section headings, these 3 sections should have a brief presentation/introduction, contextualizing what will be presented in each of them.

In terms of content, in the Abstract, the second last sentence should be removed as it is repeated in the last sentence.

In the Introduction, at the end of the first paragraph, reference is made to hybrid models (line 21), but these models are not referred to in line 17, where reference is made to time series models and machine learning models.

In line 32, the sentence seems to be incomplete.

In line 40, instead of “longer text,” it should be “longer time series”.

In line 41, the sentence is incomplete.

The Related Work section only presents 3 forecasting methods that will be compared with the proposed method. This section should contain references to works that use these methods and make similar comparisons to those proposed by the authors in the rest of the article.

In the methodology section, it would be better to start by presenting the Standard Transformer approach, shown in Figure 2, and only then move on to the Linear Transformers.

Right after Equation 2, the text following "Where" does not describe the variables in the equation.

Figures 4 and 5 need a higher zoom level.

In the Experiment section, the description of the datasets is lengthy and does not contain details about the data. It only describes details of the institutions from which the data come. The important thing would be to describe the number of time series, the number of observations, etc.

In section 4.2, lines 268 to 270 contain the words "training" and "model" repeated 3 times.

In line 272, the meaning of "data with additional difficulty" is not clear.

It may be necessary to indicate the meaning of AUC and briefly explain what this metric allows to analyze.

In the Results section, section 4.3, figures 6 to 11 have an excessive zoom level. They are too large. Figure 12 is too small.

In this section, one fact that is difficult to understand is that the methods/models used in each comparison are not always the same.

In figures 8 and 9, and in the description associated with those figures, it is necessary to present the accuracy error associated with prediction accuracy.

Figures 8 and 9 do not appear in the correct sequence.

The paragraph from lines 300 to 303 and the algorithm that appears afterwards should be moved to the methodology section.

The units in Figure 10 are not displayed properly. It should be (M).

Figure 12 does not have units on the vertical axis.

The Conclusions section contains incomplete sentences, especially in the first two paragraphs (lines 332 to 354).

The third paragraph (lines 355 to 370) is more appropriate to move to the beginning of the Conclusions section.

Author Response

Reviewer2:        

The subject of the article is current and important, and it is a topic of interest to be presented in Applied Sciences. The article has a set of recent references, adequate in both number and quality of articles.

Comments:

  1. In terms of structure, the article is well-structured. However, to avoid sections 2, 3, and 4 starting with sub-section headings, these 3 sections should have a brief presentation/introduction, contextualizing what will be presented in each of them.

Response:

Thanks to your suggestion, we have included short introductions under sections 2, 3, and 4 to contextualize what is covered in each section. In section 2, we have added: In this section, we have selected three models that are commonly used for time series prediction, which represent different kinds of models, and each section gives a short introduction to them. In section 3, we add: In this section, we mainly introduce the approach of this paper, firstly, we introduce the overview, then we introduce the replacement process of linear Transformer, the principle of multi-headed attention mechanism, and how we combine them. In section 4, we added: In this section, the main introduction is the experimental data setup, the details of the experiment, and the experimental results and analysis.

Comments:

  1. In terms of content, in the Abstract, the second last sentence should be removed as it is repeated in the last sentence.

Response:

Thank you for your reminder, we have removed the duplicate parts of the abstract.

Comments:

  1. In the Introduction, at the end of the first paragraph, reference is made to hybrid models (line 21), but these models are not referred to in line 17, where reference is made to time series models and machine learning models.

Response:

Thank you for the heads up, this was indeed an oversight on our part and we will revise this section to make it more logical already. I am going to add the hybrid model in line 17.

Comments:

  1. In line 32, the sentence seems to be incomplete.

In line 40, instead of “longer text,” it should be “longer time series”. 

In line 41, the sentence is incomplete. 

Response:

Thank you for your questions, I have reviewed our article carefully and indeed there are these problems, we have made changes in the manuscript to complete lines 32 and 41 and replace "longer text," with "longer time series" in line 40.

Comments:

5.This section should contain references to works that use these methods and make similar comparisons to those proposed by the authors in the rest of the article. 

Response:

Thanks to your suggestion, I have added references in The Related Work section to compare with their relevance.

Comments:

6.In the methodology section, it would be better to start by presenting the Standard Transformer approach, shown in Figure 2, and only then move on to the Linear Transformers. 

Response:

Thank you for your suggestion, we have adjusted the training in the manuscript, firstly introducing Standard Transformer approach, we hope our modification can satisfy you and look forward to your reply.

Comments:

7.Right after Equation 2, the text following "Where" does not describe the variables in the equation. 

Response:

Thanks to your reminder, we have added a description of the equation variables in the "Where" section after Equation 2.

Comments:

8.Figures 4 and 5 need a higher zoom level. 

Response:

Thanks to your suggestion, we have enlarged the size of Figures 4 and 5 in the manuscript to make it more clear.

Comments:

9.In the Experiment section, the description of the datasets is lengthy and does not contain details about the data. It only describes details of the institutions from which the data come. The important thing would be to describe the number of time series, the number of observations, etc. 

Response:

Thanks to your suggestion, we have added some content to the dataset to describe the number of time series and the number of observations. In our experiments, the number of time series we selected is 800 and the number of observations is 200.

Comments:

10.In section 4.2, lines 268 to 270 contain the words "training" and "model" repeated 3 times. 

Response:

Thanks to your reminder, we have rewritten lines 268 to 270 to make their presentation clearer. We changed this sentence to read: We first compare the training time of different models with the proposed model.

Comments:

11.In line 272, the meaning of "data with additional difficulty" is not clear.

Response:

Thanks for your feedback. I have modified "data with additional difficulty" to make its meaning clearer, as follows: We also compared the prediction accuracy of different models on more complex datasets. I hope that our modifications can satisfy you.

Comments:

12.It may be necessary to indicate the meaning of AUC and briefly explain what this metric allows to analyze. 

Response:

Thanks to your suggestion, we have added, below Figure 13, an explanation of the meaning of AUC and a brief description of what this metric can analyze. The additions are: AUC (Area Under the ROC Curve) is a commonly used binary classification model evaluation index, which is the area under the ROC curve whose value ranges from 0 to 1. The larger the AUC, the better the model's performance and ability to distinguish between positive and negative samples.

Comments:

13.In the Results section, section 4.3, figures 6 to 11 have an excessive zoom level. They are too large. Figure 12 is too small. 

Response:

Thanks to your suggestions, we have made figures 6 to 11 smaller and figure 12 larger to make it more comfortable to look at.

Comments:

14.In this section, one fact that is difficult to understand is that the methods/models used in each comparison are not always the same. In figures 8 and 9, and in the description associated with those figures, it is necessary to present the accuracy error associated with prediction accuracy. Figures 8 and 9 do not appear in the correct sequence.

Response:

Thank you for your question, in order to test the performance of more models, so we always use different models in each experiment, so that we can compare more comprehensively with the proposed models, where the more important models with contrast, such as LSTM and Transformer models, we have made comparisons in each experiment. We added an explanation of the accuracy errors in Figures 8 and 9, as well as in the descriptions associated with these figures, where we added: The precision error related to prediction accuracy refers to the error between the model prediction results and the real results in machine learning. Accuracy error is usually used to evaluate the model's performance. A smaller accuracy error indicates that the results predicted by the model are closer to the real results, indicating the better performance of the model.

The following are several common precision errors related to prediction accuracy:

Mean Squared Error (Mean Squared Error, MSE): MSE is one of the most common accuracy errors, the average square of the difference between the predicted and true values. The smaller the MSE, the closer the predicted result of the model is to the real value;

Mean Absolute Error (Mean Absolute Error, MAE): MAE is the average value of the absolute value of the difference between the predicted value and the true value. The smaller the MAE, the closer the predicted result of the model is to the real value;

Root Mean Squared Error (Root Mean Squared Error, RMSE): RMSE is the square root of MSE, similar to MSE but more sensitive to outliers. The smaller the RMSE, the closer the predicted result of the model is to the real value;

Mean Absolute Percentage Error (MAPE): MAPE is the average value of the absolute value of the difference between the predicted value and the true value divided by the percentage of the true value. MAPE can be used to evaluate the performance of the model at different scales; We change the order so that Figures 8 and 9 appear in the correct order.

Comments:

15.The paragraph from lines 300 to 303 and the algorithm that appears afterwards should be moved to the methodology section. 

Response:

Thanks to your suggestion, we have moved this section to the methodology section and hope that our changes will satisfy you.

Comments:

16.The units in Figure 10 are not displayed properly. It should be (M). 

Response:

Thanks to your suggestion, we have made a change in the figure and changed the unit to (M).

Comments:

17.Figure 12 does not have units on the vertical axis. 

Response:

Thanks for the reminder, I have added the units to the vertical axis.

Comments:

18.The Conclusions section contains incomplete sentences, especially in the first two paragraphs (lines 332 to 354).

Response:

Thank you for your suggestion, we have thoroughly checked the conclusion section and added the incomplete sentences, to complete them. We hope our changes will meet your requirements.

Comments:

19.The third paragraph (lines 355 to 370) is more appropriate to move to the beginning of the Conclusions section. 

Response:

Thanks to your suggestion, I have placed the third paragraph section at the beginning.

Reviewer 3 Report

This paper presents a model based on multiplexed attention mechanism and linear Transformer to predict financial time series.

1- The abstract of the paper is poorly written with some sentences repeated, thus it would be beneficial to rephrase it in a way that effectively conveys the main ideas of the paper.

2- It is important to clearly discuss the limitations of the proposed model.

3- The clarity of the paper's figures, particularly figures 1, 2, 4, and 5, is inadequate. To improve comprehension, please provide clearer figures.

4- In addition to the main findings, the paper should discuss future work in the conclusion that has not been covered in other sections of the paper.

5- It would be appreciated if the Conclusion and Discussion sections were divided into separate sections.

Author Response

reviewer3:            Comments and Suggestions for Authors

This paper presents a model based on multiplexed attention mechanism and linear Transformer to predict financial time series.

Comments:

1.The abstract of the paper is poorly written with some sentences repeated, thus it would be beneficial to rephrase it in a way that effectively conveys the main ideas of the paper. 

Response:

Thank you for your suggestion, we removed the repetitive parts of the abstract and re-reviewed the content of the abstract, we re-wrote the content of the abstract to effectively convey the main ideas of the paper, we hope our changes will satisfy you,the re-written abstract is as follows:Financial time series prediction has been an important topic in deep learning, and the prediction of financial time series is of great importance to investors, commercial banks and regulators. In this paper, we propose a model based on multiplexed attention mechanism and linear transformer to predict financial time series. The linear transformer model has faster model training efficiency and long time forecasting capability. Using a linear transformer reduces the complexity of the original transformer and preserves the multiplexed attention mechanism in the decoder. The results show that the proposed method can effectively improve the prediction accuracy of the model, increase the inference speed of the model, and reduce the amount of operations, which has new implications for the prediction of financial time series.

Comments:

2.It is important to clearly discuss the limitations of the proposed model. 

Response:

Thanks to your feedback, we have added the limitations of the proposed model and future perspectives to the conclusion section. We hope you will be satisfied with our modifications. Linear Transformer models based on multi-headed attention mechanisms are highly effective in various natural language processing tasks, including language modeling, machine translation, and text classification. However, they also have some limitations and potential directions for future research. One limitation of linear Transformer models is that they can be computationally expensive, especially when dealing with long time series sequences. This can slow training and inference times, which can be a problem in real-time applications. To address this, researchers are exploring ways to optimize the computation of attention weights, such as using scant attention or pruning attention heads. Another limitation is that linear Transformers must be better suited for tasks requiring reasoning or inference, such as question-answering or dialogue generation. This is because they lack explicit mechanisms for representing and manipulating structured knowledge. Researchers are exploring ways to incorporate external knowledge sources, such as knowledge graphs or ontologies, into the model architecture to address this. In terms of future directions, one promising area of research is the development of hybrid models that combine the strengths of linear Transformers with other neural network architectures, such as convolutional neural networks or graph neural networks. Another direction is the exploration of unsupervised pre-training techniques, such as generative pre-training or contrastive learning, to improve the efficiency and effectiveness of linear Transformer models.

Comments:

3.The clarity of the paper's figures, particularly figures 1, 2, 4, and 5, is inadequate. To improve comprehension, please provide clearer figures. 

Response:

Thanks to your feedback, I submitted clearer images in the text to improve the readability of the article.

Comments:

4.In addition to the main findings, the paper should discuss future work in the conclusion that has not been covered in other sections of the paper. 

Response:

Thanks to your feedback, we have included in the summary section of the article a discussion of future work not covered in other parts of the paper. We hope that our changes will satisfy you. The additions are as follows: Linear Transformer models based on multi-headed attention mechanisms are highly effective in various natural language processing tasks, including language modeling, machine translation, and text classification. However, they also have some limitations and potential directions for future research. One limitation of linear Transformer models is that they can be computationally expensive, especially when dealing with long time series sequences. This can slow training and inference times}, which can be a problem in real-time applications. To address this, researchers are exploring ways to optimize the computation of attention weights, such as using scant attention or pruning attention heads. Another limitation is that linear Transformers must be better suited for tasks requiring reasoning or inference, such as question-answering or dialogue generation. This is because they lack explicit mechanisms for representing and manipulating structured knowledge. Researchers are exploring ways to incorporate external knowledge sources, such as knowledge graphs or ontologies, into the model architecture to address this. In terms of future directions, one promising area of research is the development of hybrid models that combine the strengths of linear Transformers with other neural network architectures, such as convolutional neural networks or graph neural networks. Another direction is the exploration of unsupervised pre-training techniques, such as generative pre-training or contrastive learning, to improve the efficiency and effectiveness of linear Transformer models.

Comments:

5.It would be appreciated if the Conclusion and Discussion sections were divided into separate sections. 

Response:

Thank you for your suggestion, this does make the content of our manuscript clearer and we have divided the conclusion and discussion sections into separate chapters to increase the readability of our manuscript.

Round 2

Reviewer 1 Report

The revised version is satisfactory and is now suitable for publication. However, I some minor comments

1) Page 348, "the proposed model s more" should be corrected to "the proposed model is more".

2) Page 331, "the proposed model s, we selected" should be corrected to  "the proposed model, we selected"

3) The Discussion section should be placed before the Conclusions section.

Author Response

Reviewer1:

                              Comments and Suggestions for Authors

The revised version is satisfactory and is now suitable for publication. However, I some minor comments

Comments:

  1. Page 348, "the proposed model s more" should be corrected to "the proposed model is more".

Response:

Thank you for providing feedback on our manuscript. Per your suggestion, we changed "the proposed model s more" to "the proposed model is more." I hope our modification can satisfy you.

Comments:

  1. Page 331, "the proposed model s, we selected" should be corrected to  "the proposed model, we selected"

Response:

Thank you for providing feedback on our manuscript.

"the proposed model s we selected" was changed to "should be corrected to "according to your suggestion; we will hope that our modification can satisfy you.

Comments

3.The Discussion section should be placed before the Conclusions section.

Response:

Thank you for your suggestions and reminders. This is indeed a place that we need to consider carefully. We have swapped The Discussion and Conclusions sections to make our manuscript more reasonable. We hope our revisions will satisfy you and await your reply.

Reviewer 2 Report

The authors updated the paper following most of the suggestions provided by reviewers.

Even though, there are one point in the new material that should be revised:

Section 6 (Discussion) should appear before Section 5 (Conclusions).

While there was an improvement in the English language compared to the previous version, it would still be necessary to conduct a comprehensive revision once all revisions have been completed.

Author Response

Reviewer2:                Comments and Suggestions for Authors

Comments:

The authors updated the paper following most of the suggestions provided by reviewers.

Even though, there are one point in the new material that should be revised:

Section 6 (Discussion) should appear before Section 5 (Conclusions).

While there was an improvement in the English language compared to the previous version, it would still be necessary to conduct a comprehensive revision once all revisions have been completed.

Response:

Thank you for your suggestions and reminders. This is indeed a place where we could have thought more carefully. We switched The Discussion and Conclusions sections to make our manuscript more reasonable. Then, we revised our manuscript again. Language Hope that our modification can satisfy you and look forward to your reply.

Reviewer 3 Report

The authors have considered all the comments provided. With the necessary revisions made, the paper is now ready for acceptance. I only have one additional comment, which is to replace "Ours" in the figure legend with "The Proposed Model".

Author Response

Reviewer 3:Comments and Suggestions for Authors

The authors have considered all the comments provided. With the necessary revisions made, the paper is now ready for acceptance. I only have one additional comment, which is to replace "Ours" in the figure legend with "The Proposed Model".

Response:

Thank you for your reminder. In response to your suggestion, we changed all "Ours" in the legend to "The Proposed Model" to correspond to the information in the text.